# New Techniques in Hemorrhoidal Disease but the Same Old Problem: Anal Stenosis

**DOI:** 10.3390/medicina58030362

**Published:** 2022-03-01

**Authors:** Sezai Leventoglu, Bulent Mentes, Bengi Balci, Halil Can Kebiz

**Affiliations:** 1Department of General Surgery, Faculty of Medicine, Gazi University, Cankaya, Ankara 06560, Turkey; 2Department of Surgery, Proctology, Ankara Memorial Hospital, Cankaya, Ankara 06520, Turkey; bulent.mentes@memorial.com.tr; 3Department of General Surgery, Ankara Oncology Training and Research Hospital, Yenimahalle, Ankara 06105, Turkey; bengibalci@gmail.com; 4Faculty of Medicine, Gazi University, Cankaya, Ankara 06560, Turkey; hckebiz@gmail.com

**Keywords:** anal stenosis, hemorrhoidectomy, diamond flap, house advancement flap, rhomboid flap, Y-V flap

## Abstract

Anal stenosis, which develops as a result of aggressive excisional hemorrhoidectomy, especially with the stoutly use of advanced technologies (LigaSure^®^, ultrasonic dissector, laser, etc.), has become common, causing significant deterioration in the patient’s quality of life. Although non-surgical treatment is effective for mild anal stenosis, surgical reconstruction is unavoidable for moderate to severe anal stenosis that causes distressing, severe anal pain, and inability to defecate. The problem in anal stenosis, unlike anal fissure, is that the skin does not stretch as a result of chronic fibrosis due to surgery. Therefore, the application of lateral internal sphincterotomy does not provide satisfactory results in the treatment of anal stenosis. Surgical treatment methods are based on the use of flaps of different shapes and sizes to reconstruct the anal caliber and flexibility. This article aims to summarize most-used surgical techniques for anal stenosis regarding functional and surgical outcomes.

## 1. Introduction

Anal stenosis (AS) is defined as anatomical or functional narrowing of the anal canal, which can result from inflammatory bowel diseases, radiation therapy, congenital malformations, or excisional hemorrhoidectomy [1,2]. The anatomical AS is related to the increased fibrous scar tissue forming, which disables stretching of the anal canal [3]. The leading cause of the anatomical AS is excisional hemorrhoidectomy that is often the chosen treatment for grade III and IV hemorrhoidal disease [4].

The incidence of AS is reported to be as high as 5%, and patients usually present with burdensome symptoms such as severe constipation, outlet obstruction, and anal pain, which cannot be alleviated with stool softeners or dietary changes [3,5,6]. The diagnosis can be made during rectal examination by visualizing the scar tissue and the extension of the anal stricture, localized or circumferential. 

Milsom and Mazier described a classification system for the postsurgical AS that defines the treatment options based on the severity and the level of the stricture (Table 1) [7]. Non-operative management, including mechanical dilatation, fiber supplements, and stool softeners, may achieve good results in selected cases with mild AS [8,9]. However, operative treatment is inevitable for moderate AS refractory to non-operative management and severe AS.

This article aims to review the operative treatment methods regarding functional results, postoperative care, and complications.

## 2. Surgical Techniques

Several flap techniques have been described for the treatment of AS, and they can mainly be classified as advancement, island (adjacent tissue transfer), or rotational flaps [10] (Table 2). These techniques are based on delivering the more pliable anoderm into the anal canal to replace the scar tissue [1]. Depending on the extension of the stricture into the anal canal and the presence of adequate perianal skin, one of those techniques can be performed unilaterally or in several quadrants of the anal verge.

On the other hand, there is significant heterogeneity in the reported studies in terms of sample sizes (ranging from 4 to 149 patients), subjective assessment of functional outcomes (good-fair-poor) without the use of standardized scoring systems, and the evaluation of healing [11,12,13,14,15,16,17,18] (Table 3). Those terms result in choosing the surgical technique based on the surgeon’s familiarity rather than the patient’s clinical features.

***a.*** 
**
*Mucosal advancement flap*
**


The mucosal advancement flap is mainly preferred for the treatment of mid-level AS. Good functional outcomes have been reported by a couple of studies that include different samples sizes [2,19]. Rakhmanine et al. reported outcomes of 95 patients in a retrospective study, and the overall complication rate was found to be 3% [20].


Technical notes of Mucosal advancement flap:


The technique starts with the excision of the scar tissue; then, a transverse incision is made proximally to the dentate line. The rectal mucosa is dissected to the level of submucosa and advanced to the anal canal to cover the excised stricture area. The exterior wound is preferred to be left open to minimize the ectropion formation [1]. 

***b.*** 
**
*House flap*
**


The house flap is recommended if the stenosis extends from the dentate line to the perianal skin. The creation of a wide-based flap increases the anal canal diameter along its length and allows primary closure of the donor site [1]. Alver et al. reported complete healing in all 28 patients [21], whereas Sentovich et al. demonstrated healing rates of 89% with a median follow-up of 28 months [22]. A prospective randomized study revealed a clinical improvement rate of 90% with the house flap when compared to rhomboid flap (80%) and Y-V anoplasty (65%) during 1-year follow-up [23].


Technical notes of House flap:


The patient is placed in the prone jack-knife position. The longitudinal incision is made from the dentate line to the end of the stenosis. Then the flap is designed as in the house shape, with the length of the “walls” of the flap corresponding to the length of the incision and the “roof” of the flap reaching the healthy perianal skin (Figure 1 and Figure 2). The flap is dissected to the depth of the ischiorectal fat to advance it into the anal canal without tension and to preserve the vascular pedicle [1,21].

***c.*** 
**
*Diamond flap*
**


The diamond flap was first described by Caplin and Kodner in 1986 and has been performed for the treatment of moderate and severe AS [24]. Although the final target for anal caliber has not been standardized, Gulen et al. reported a clinical success rate of 88.9% with an eventual anal caliber of 25 to 26 mm in 18 consecutive patients. After 12-months of follow-up, the obstructed defecation syndrome scores were found to be significantly improved [25]. 


Technical notes of Diamond flap:


The patient is placed in the prone jack-knife position. The incision is made on the scar tissue longitudinally until it reaches the dentate line. The anal caliber is recommended to be checked during this step, and the external sphincter should be spared [26]. The diamond flap is designed as adjacent to the diseased anal canal. The flap is dissected out together with its vascular pedicle and advanced into the anal canal. The flap should be tension-free to avoid any postoperative wound complications (Figure 3, Figure 4 and Figure 5).

***d.*** 
**
*Y-V flap/V-Y flap*
**


The Y-V advancement flap is performed for low and localized strictures below the dentate line, whereas the V-Y flap is used for mild to severe stricture at the dentate line [1]. Maria et al. reported 29 patients who had Y-V anoplasty in a comparative study with diamond flap, with a healing rate of 90% and an ischemic contracture of the leading edge of the flap and wound dehiscence in two patients [27]. The authors suggested that the diamond flap is a more reliable technique due to the reduced tension in the suture line and the better blood supply of the flap. Farid et al. observed less clinical improvement (65%) and higher ischemic wound complications (15%) with this technique [23].


Technical notes of Y-V flap/V-Y flap:


Initially, the vertical limb of the Y is performed to the area of stricture. The incision is extended to the perianal skin in two directions creating a V shape. Then, the tip of the V is advanced to the vertical limb of the Y incision. It can also be done in either a lateral position or just in the posterior midline. The main disadvantage regarding this technique is the tip of the V flap being prone to ischemic necrosis [10].

For the V-Y advancement flap, the first few steps are the same as the Y-V flap, but in this procedure, the wider base of the triangular Y flap is sutured to the dentate line. Then the perianal skin is approximated with sutures longitudinally behind the V shape to form the Y shape [1].

***a.*** 
**
*Rhomboid flap/Modified rhomboid flap*
**


The rhomboid flap has been modified in terms of the flap size to be adjusted to each patient. Sloane et al. observed good functional results in a case series including nine patients, eight of whom underwent bilateral rhomboid flap with complete resolution of symptoms [28]. The modified rhomboid flap is demonstrated as a safe and suitable technique for the treatment of moderate and severe AS. Gallo et al. reported 0% recurrence rate and 96% success rate in a study with 50 consecutive patients, and significant improvement in the obstructed defecation syndrome scores and the quality of life were observed at 12 months [29]. The mean anal caliber was found to be 24 mm and significantly different compared to the preoperative measurement.


Technical notes of Rhomboid flap/Modified rhomboid flap:


The size of the rhomboid flap can vary between 8 to 9 cm in length and 5 cm in width at its largest point. A minimal left/right internal sphincterotomy can be performed by paying careful attention to the external sphincter. Then, the flap is relocated in the anal canal and fixed with a single-layer suture of absorbable stitches to the distal rectum. The skin sutures should be adequately spaced to avoid excessive tension with subsequent ischemia [29].

***a.*** 
**
*U-flap*
**


The U-flap technique is mainly used for the treatment of AS with mucosal ectropion. The disadvantage of this technique is that the donor site is left open [1]. Pearl et al. reported a good clinical result of 92% in 25 patients during a mean follow-up of 19 months [30].


Technical notes of U-flap:


The procedure starts with the excision of the ectropion; then, a U-shaped incision is made at the adjacent perianal skin. The U flap is advanced into the anal canal to cover the wide defect resulting from the excised area of ectropion.

***b.*** 
**
*Rotational S-plasty*
**


Rotational S-plasty was first described by Ferguson in 1959 for the repair of the whitehead deformity of the anus [31]. Corman et al. in 1976 adapted this technique for the treatment of AS [32]. This technique enables covering large areas of skin with an adequate blood supply. Gonzalez et al. revealed that 94% of patients had good results during a mean follow-up of 18 months [33].


Technical notes of Rotational S-plasty:


The technique starts with outlining the semicircular incision in the perianal skin, with a length of 12 to 13 cm. The base of the incision should not be shorter than its length. Then skin incision is done to the layer of subcutaneous fat lobules to preserve the blood supply of the skin flaps. The flap is then rotated and fixed to the mucosa by interrupted sutures [34].

## 3. Postoperative Care and Complications

Patients are usually discharged on postoperative day 1, with recommendations of daily sitz baths or showers for comfort and hygiene. Prophylactic antibiotics with metronidazole and ciprofloxacin or cephalosporins can be continued in the postoperative setting. The postoperative complications encountered commonly are urinary retention, wound dehiscence, wound infection, flap ischemia, and bleeding [13,19,20,27].

## 4. Conclusions

The AS due to overzealous hemorrhoidectomy is an entirely preventable disease when performed under skilled and experienced hands [35]. With better knowledge of the anorectal anatomy and delicate treatment to the anal tissue, the complications of anorectal surgery, such as anal stenosis, can be reduced. Considering the fact that with today’s technological evolution, there are many alternative techniques such as doppler-guided hemorrhoidal artery ligation and stapled hemorrhoidopexy, and excisional hemorrhoidectomy may not be the only option for many patients [36].

Nevertheless, anoplasty techniques should be in the armamentarium of colorectal surgeons. The basis of these techniques includes excision or incision of the scar tissue, preparation of the flap with careful attention to the vascular supply, advancement of the flap, and fixating the flap without tension into the anal canal. Following these steps carefully would result in a significant decrease in anoplasty complications.

As for the better technique regarding functional outcomes, we need more high-quality studies objectively evaluating patients’ quality of life and functional outcomes using standardized scoring systems and questionnaires.

## Figures and Tables

**Figure 1 medicina-58-00362-f001:**
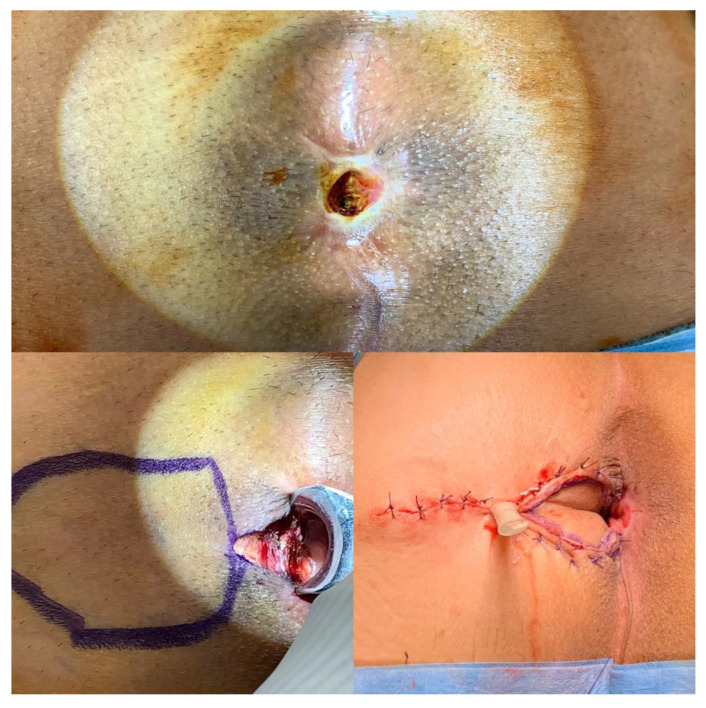
The house flap anoplasty is shown in a patient with severe anal stenosis.

**Figure 2 medicina-58-00362-f002:**
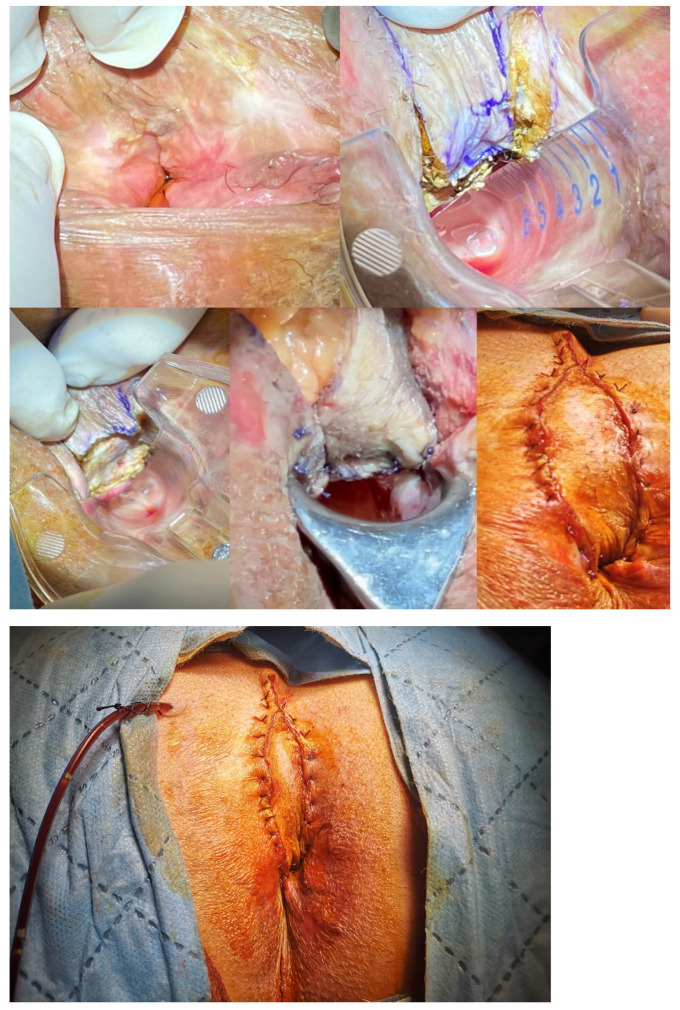
A tailored house flap anoplasty in a patient with chronic unhealing wound in the posterior anal canal.

**Figure 3 medicina-58-00362-f003:**
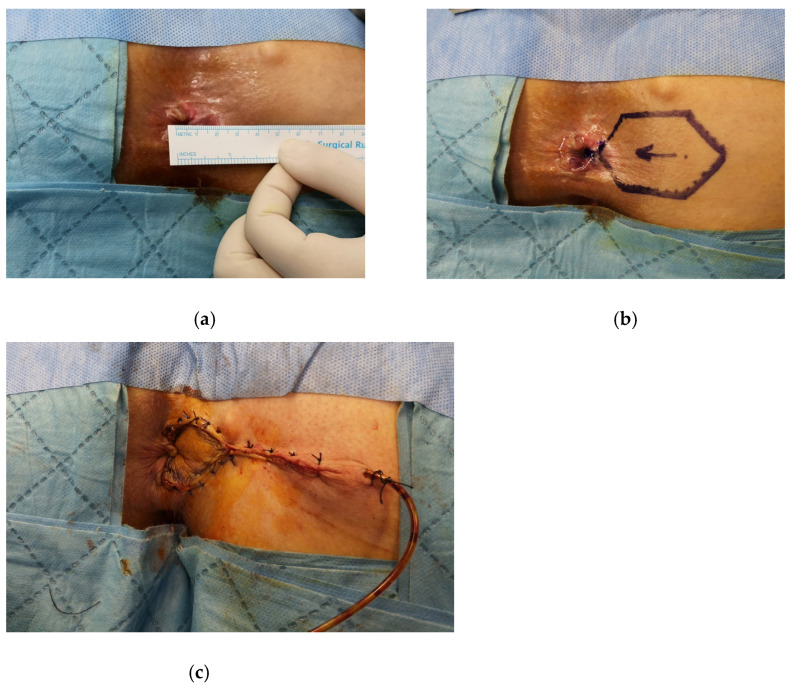
The diamond flap anoplasty is shown in a patient with severe anal stenosis, (**a**) the anal caliber is measured as 10 mm, (**b**) the right-sided anoplasty is decided according to the location of scar tissue in the anal verge, (**c**) the anoplasty is performed and the negative pressure drain is placed.

**Figure 4 medicina-58-00362-f004:**
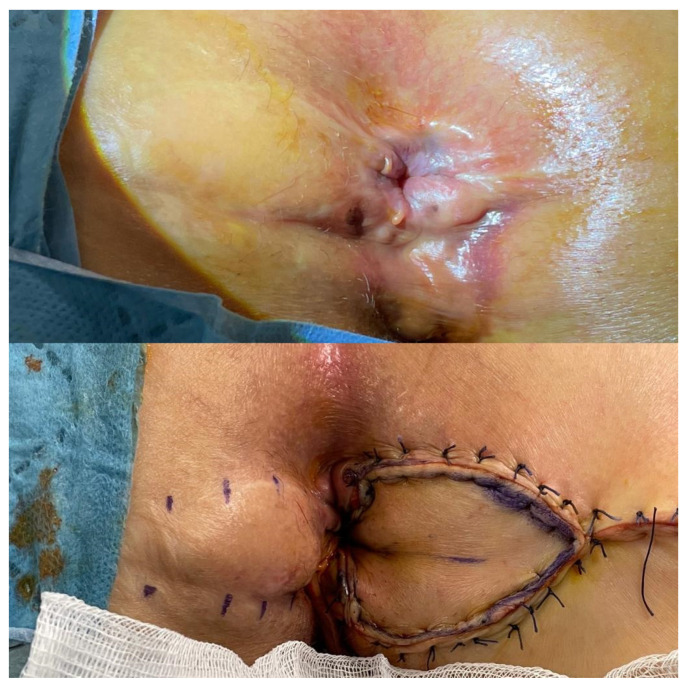
Contralateral diamond flap anoplasty in a patient with recurrent anal stenosis.

**Figure 5 medicina-58-00362-f005:**
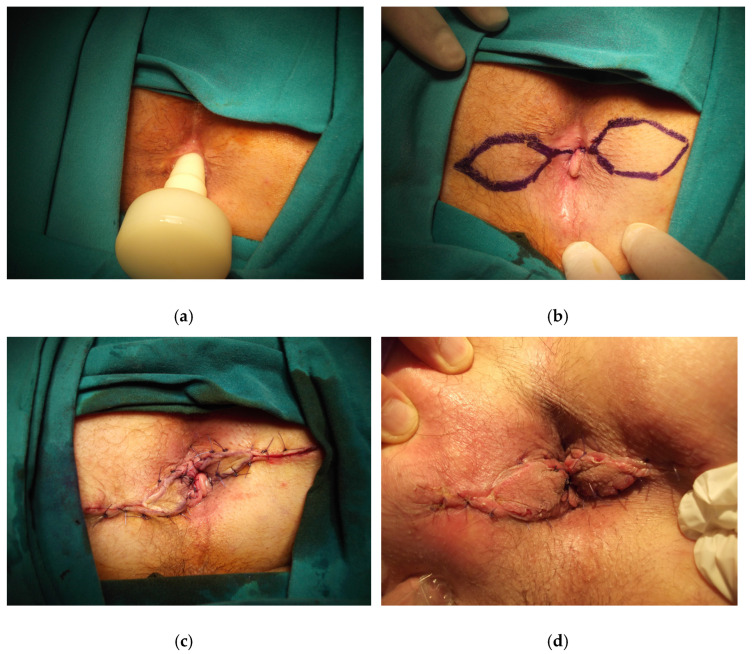
Bilateral diamond flap anoplasty in severe anal stenosis is shown, (**a**) the anal caliber is measured using an anal calibrator, (**b**) the technique is drawn, (**c**) the postoperative image after anoplasty, (**d**) the image in postoperative 1st week.

**Table 1 medicina-58-00362-t001:** Classification of anal stenosis by Milsom and Mazier [7].

**Classification based on the severity**
Mild: Tight anal canal can be examined by a well-lubricated index finger or a medium Hill-Ferguson retractor.
Moderate: Forceful dilatation is required to insert either the index finger or a medium Hill-Ferguson retractor.
Severe: Neither the little finger nor the small Hill-Ferguson retractor can be inserted unless a forceful dilatation is employed.
**Classification based on the level of stenosis**
Low: Distal anal canal at least 0.5 cm below the dentate line
Middle: 0.5 cm proximal to 0.5 cm distal to the dentate line
High: Proximal to 0.5 cm above the dentate line

**Table 2 medicina-58-00362-t002:** Common surgical techniques used for treating anal stenosis.

Surgical Technique	Indications	Advantages	Disadvantages
Mucosal advancement flap	Middle or high mild anal stenosis	-	The risk of ectropion unless the wound is left open
House flap	Moderate to severe anal stenosis	Provides adequate extension in the anal canal diameter	-
Diamond flap	Moderate to severe anal stenosis	Covers the defect in the anal canal while sparing the sphincter complex	-
Y-V flap/V-Y flap	Mild to moderate anal stenosis	-	Flap’s tip prone to ischemia and lacks sufficient extension of the anal canal diameter
Rhomboid flap/Modified rhomboid flap	Moderate to severe anal stenosis	Enables a tailored-anoplasty in different sizes	-
U-flap	Excising the mucosal ectropion	-	Leaving the wound open results in delay of recovery
Rotational S-plasty	Moderate to severe anal stenosis	Provides a large tissue rotation without compromising vascular supply	-

**Table 3 medicina-58-00362-t003:** Several studies on functional and surgical outcomes after anoplasty.

Authors	Study Method	Total N of Included Patients	Indications for Anoplasty (N of Patients)	Surgical Techniques	Functional Outcomes	Surgical Outcomes (N of Patients)	Mean Follow-Up (Months)
Rakhmanine et al. [20]	Retrospective	95	Hemorrhoidectomy (35) Chronic anal fissure (10) Perianal fistula (4) Anal carcinoma (1) Various (10)	Mucosal advancement flap	Reported as good in 74 patients and as poor in 8 patients	Abscess (1) Seepage of liquid stool (2)	50
Alver et al. [21]	Retrospective	28	Chronic anal fissure (14) Anal stenosis (8) Rectovaginal fistula (1) Perianal fistula (3) Anal carcinoma (1) Obstetric injury (1)	House flap	Reported as good in 8 patients with anal stenosis	Wound dehiscence (3) Recurrence of rectovaginal fistula (1)	26
Sentovich et al. [22]	Retrospective	29	Anal stenosis (21) Ectropion (4) Bowen’s disease (2) Key-hole deformity (2) Perianal fistula (1)	House flap	Reported as good in 26 patients and as poor in 3 patients	Donor-site separation (14) Urinary retention (8) Sepsis (4)	28
Farid et al. [23]	Prospective-randomized	60	Anal stenosis (60)	Rhomboid flap/Y-V flap/House flap	Better anal caliber increase and improvement in GI-QLI score with house-flap		12
Gulen et al. [25]	Retrospective	18	Anal stenosis (18)	Diamond flap	Significant increase in anal caliber and improvement in ODS score	Wound dehiscence (4)	35
Maria et al. [27]	Comparative	42	Anal stenosis (42)	Diamond flap/Y-V flap	Reported as good in 89% of patients with Y-V flap, and 100% with diamond flap	Wound dehiscence (1) Ischemia in tip of the flap (1)	24
Sloane et al. [28]	Retrospective	9	Anal stenosis (9)	Rhomboid flap	Reported as significant improvements in 9 patients	Single quadrant stenosis (1)	12
Gallo et al. [29]	Retrospective	50	Anal stenosis (50)	Modified rhomboid flap	Significant increase in anal caliber and improvement in ODS and CCI score	Ischemia of donor site (1) Wound dehiscence (2)	97
Pearl et al. [30]	Retrospective	25	Anal stenosis (20) Ectropion (5)	Island flap (U-shaped and Diamond-shaped)	Reported as excellent in 64% of patients and good in 25% of patients	-	19
Gonzalez et al. [33]	Comparative	17	Anal stenosis (13) Perianal fistula (2) Key-hole deformity (1) Chronic anal fissure (1)	Rotational S-plasty/Advancement flap	Reported as good in 16 patients	Sepsis (1)	18

Abbreviations: N, number; GI-QLI, gastrointestinal quality of life inventory; ODS, obstructed defecation syndrome; CCI, Cleveland Clinic Incontinence.

## Data Availability

This study did not report any new research data.

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
