# Peer review of "New Techniques in Hemorrhoidal Disease but the Same Old Problem: Anal Stenosis"

_medicina, 2022, doi:10.3390/medicina58030362_

Round 1

Reviewer 1 Report

Very comprehensive technical paper on a significant morbidity issue following hemorrhoidectomy. Photos of the anal canal reconstruction may need better magnification.

Author Response

Thank you for your interest in our paper and your kind reviews. As you may know, surgical photos are difficult in technicality, especially when it is taken by a non-professional person. Unfortunately present photos are the best ones we have had.

Reviewer 2 Report

The revision of various methods for managing anal stenosis is urely interesting. The structure adheres to the standard of the narrative review without however challenging itself to following a set of guidelines; this creates a vacuum of scientific validity. The article is comparabale to a friendly lesson and lacks the support of a stricter EBM background. Surely a couple of tables with a bit of data regading cited studies could help. Nonetheless the article is readable, the English is correct even if structure must be revised: a lot of spaces after commas must be added. Readibility could be improved with better paragraph formatting. The included table (Table1) is a bit confusing and can be hugely improved. The selection of images is high quality and understandable.

The real problem of the article is the complete absence of a "material and methods" section, that nullifies every chance to be scientifically acceptable, especially if the goal is to give a review of literature. Adding an in-depth but short section can hugely imrpve the standards of this work.

Author Response

Thank you for your interest in reviewing our manuscript. We aimed to write a review article focusing on the technicalities of surgical techniques, not systematic review or literature review. Since it is not a systematic review article, material and methods section is not required. 

However, in line with your recommendations, we have added two additional tables; first one is to summarize surgical techniques in terms of indications, advantages and disadvantages. The second added table is to summarize several cited studies regarding study method, number of included patients, investigated technique, and functional and surgical outcomes of the study.

We believe with these modifications, hope you will find this article a valuable addition to the current literature.

Reviewer 3 Report

The authors reviewed the details of anal stenosis, including the etiologies, treatments and outcomes. It's necessary to summarized the treatment options and different clinical outcomes with references into a table.  In this way, the readers can understand more easily. 

Author Response

Thanks for your interest in reviewing our manuscript.  Upon your recommendations we have added two tables; first one is to summarize surgical techniques and the other one is to summarize cited articles in terms of study method, number of included patients, surgical intervention and outcomes of the study.

Hopefully, with these modifications, you will find this article a valuable addition to the current literature.

Round 2

Reviewer 2 Report

The changes made to the article are welcome and improve its readibility.

Paragraph structure may be further improved, as the section describing the technique of each of intervention is still confusing.

The real problem is that the article aims to "review the operative treatment methods regarding functional results, postoperative care, and complications".

Thus this is no simple technical review, and even if not systematic, it requires replicability; A material and methods sections could add much with little effort, even because is practically done with the added tables.

If that is not possible the scope of the review should change and the article would be little more than personal experience with few supporting articles.

Author Response

Upon your recommendations, the scope of the article has been changed (reduced) in the abstract. In order to clear any confusion regarding the paragraph structure, titles of technical notes for each technique have been rewritten.

Reviewer 3 Report

The authors could separate advantage/disadvantage into two columns in Table 2 to make the readers easily understand the differences between these methods. Other parts were well-written and informative.

Author Response

Upon your suggestion, the last column in table 2 was separated into two columns, as advantages and disadvantages.